# Comparison of the Effectiveness of Lifestyle Modification with Other Treatments on the Incidence of Type 2 Diabetes in People at High Risk: A Network Meta-Analysis

**DOI:** 10.3390/nu11061373

**Published:** 2019-06-19

**Authors:** Kazue Yamaoka, Asuka Nemoto, Toshiro Tango

**Affiliations:** 1Teikyo University Graduate School of Public Health, 2-11-1 Kaga, Itabashi-ku, Tokyo 173-8605, Japan; anemoto@med.teikyo-u.ac.jp (A.N.); tango@medstat.jp (T.T.); 2Center for Medical Statistics, 2-9-6, Shiodome Italia-gai Siodome First 4F, Higashi Shinbashi, Minato-ku, Tokyo 105-0021, Japan

**Keywords:** systematic review, network meta-analysis, randomized controlled trial, high risk of type 2 diabetes, lifestyle modification, diet, medication

## Abstract

Background: Many clinical trials have been conducted to verify the effects of interventions for prevention of type 2 diabetes (T2D) using different treatments and outcomes. The aim of this study was to compare the effectiveness of lifestyle modifications (LM) with other treatments in persons at high risk of T2D by a network meta-analysis (NMA). Methods: Searches were performed of PUBMED up to January 2018 to identify randomized controlled trials. The odds ratio (OR) with onset of T2D at 1 year in the intervention group (LM, dietary, exercise, or medication) versus a control group (standard treatments or placebo) were the effect sizes. Frequentist and Bayesian NMAs were conducted. Results: Forty-seven interventions and 12 treatments (20,113 participants) were used for the analyses. The OR in the LM was approximately 0.46 (95% CI: 0.33 to 0.61) times lower compared to the standard intervention by the Bayesian approach. The effects of LM compared to other treatments by indirect comparisons were not significant. Conclusions: This meta-analysis further strengthened the evidence that LM reduces the onset of T2D compared to standard and placebo interventions and appears to be at least as effective as nine other treatments in preventing T2D.

## 1. Introduction

The prevalence of diabetes has been increasing worldwide and the accompanying increase in the prevalence of diabetes-related complications and the occurrence of diabetes are likely to have a substantial impact on healthcare costs [1]. Not only medical treatments, but major changes in lifestyle factors to prevent diabetes, such as diet and physical exercise, will also be needed [2]. Research on the protective mechanisms associated with physical activity, healthy eating patterns with specific food components, anti-inflammatory strategies, or weight reduction via low calorie diets should be warranted. Several meta-analyses have been performed to examine the effects in preventing type 2 diabetes of treatments involving both lifestyle modification and medications [3,4], dietary supplements [5], exercise (physical activity) [6], herbal medication [7], and numerous drugs [8]. Previously, the authors addressed the issue of lifestyle modification and examined its effects on preventing type 2 diabetes [9] as well as metabolic syndrome [10]. To cope with prevention of type 2 diabetes at the individual patient’s level, a coordinated approach like lifestyle modification that integrates the beneficial effects of education and support by health care professionals is important. Various therapies have been applied with differing results. Therefore, it might be beneficial to clarify whether lifestyle modifications would be effective for preventing the onset of type 2 diabetes compared to other proposed treatments from the viewpoint of health policy management.

Thus far, many clinical trials have been conducted to verify the effects of interventions for the prevention of type 2 diabetes using different treatments and outcomes as described above. Network meta-analyses (NMAs) incorporate correlations among interventions and have advantages compared with a univariate (pair-wise) meta-analysis of each intervention separately [11]. Specifically, for comparisons of mixed treatments or comparisons of multiple treatments in meta-analyses, an NMA expands the scope of a conventional pair-wise meta-analysis by simultaneously analyzing both direct comparisons of interventions within randomized controlled trials (RCTs) and indirect comparisons across trials based on a common comparator (e.g., placebo or some standard treatment) [12]. Furthermore, recent developments in Bayesian NMA provide a framework for direct and indirect evidence [13]. It may be helpful to assess the effectiveness of lifestyle modifications to prevent the onset of type 2 diabetes by comparison with other treatments including medications in order to clarify which aspects and what degree of lifestyle modification are superior to prevent the onset of type 2 diabetes.

This study aimed to compare the effectiveness of lifestyle modifications to other treatments for patients at high risk of type 2 diabetes by an NMA. Results of our analyses revealed important information for future treatment of those at high risk of type 2 diabetes.

## 2. Materials and Methods 

### 2.1. Study Design, Search Strategy, and Information Sources

The study design was a systematic review of published literature and a network meta-analysis of data from each selected study. This study was not registered, and there is no information on the study protocol on any web site. The study question was whether a lifestyle modification program is effective for preventing the onset of type 2 diabetes in adults at high risk of type 2 diabetes compared with other treatments. PUBMED data from inception until January 2018 were searched to identify relevant literature restricted to the English language (at least the abstract). Free text terms, Medical Subject Headings (MeSH) terms, Clinical Queries, and combinations of words were used for search terms. One investigator (KY) performed this literature search (search terms are shown in Appendix A).

### 2.2. Study Selection

Studies reporting the onset of type 2 diabetes (odds ratio, relative risk, or hazard ratio) with a follow-up of at least 6 months were examined. In total, 10 interventions (lifestyle interventions (L), dietary intervention alone (D), exercise intervention alone (E), orlistat (O), metformin (or flumamine) (M), acarbose and voglibose (A), pioglitazone and rosiglitazone (T), pitavastatin (PI), glipizide (SU), and herbal medications (H)) were treated as independent interventions. Control interventions were standard (conventional education or usual treatment) (S) or placebo (P). Specific dietary programs, such as the Mediterranean diet, were treated as dietary interventions. Details of the inclusion and exclusion criteria are shown in Appendix A. We excluded the RCT for valsartan therapy from the meta-analysis because of reported adverse events and because it is scarcely used nowadays. The grouping of these interventions was made through discussion with a pharmacologist who is one of the investigators (AN).

In the systematic review process, if results of one study were reported in more than one publication, we selected the report for the analysis that included the most information at the one-year follow-up. Furthermore, if a study reported results of interventions for high-risk patients in a stratified analysis, we treated that part of the report as an independent RCT study. When a trial had a combination treatment of two or more of the 12 treatments described, we excluded the combination arm from the analysis. Two investigators (KY, AN) independently assessed eligibility.

### 2.3. Data Extraction and Risk of Bias within Individual Studies

Participants in the examined studies were adults who were diagnosed as being at high risk of type 2 diabetes according to definitions of 2-hours plasma glucose level (2hPG) (over 7 mmol/L), impaired fasting glycaemia (IFG), or impaired glucose tolerance (IGT), or other guidelines. In selecting studies for this meta-analysis, the proportion of patients with onset of type 2 diabetes during the end of the study follow-up (≥6 months, mainly 1 year) was used and odds ratio with onset of type 2 diabetes were primarily examined in order to assess the strength of the effects. Although the maintenance of long-term control is important for the prevention of type 2 diabetes, the reason why we selected results at the 1-year study follow-up duration was to reduce duration bias. Two investigators (KY,AN) independently extracted data, and one investigator confirmed the results.

As for the risk of bias in individual studies, the quality of the studies was assessed based on how the studies had minimized bias and error in their methods following the risk of bias assessment used in the Cochrane review [14]. The assessment involved using the Cochrane collaborations tool for assessing risk of bias. The tool consists of six areas, shown in the Appendix A, where bias could possibly be introduced, and judgement could be made to assess if bias was introduced or not. Study quality and risk of bias were descriptively reported.

### 2.4. Statistical Analysis

A univariate (pair-wise) meta-analysis (UMA) and NMAs (frequentist and Bayesian models with consistency and inconsistency models) [11,13,15] were performed to estimate the odds ratio of lifestyle modification (L) on the prevention of type 2 diabetes compared to the other interventions (D, E, O, A, M, T, PI, G, H, P, and S). We qualified heterogeneity between studies with the I^2^ statistic. Evidence of inconsistency was tested using the global test for inconsistency based on the design-by-treatment inconsistency model [16]. The Bayesian NMAs were conducted within a Bayesian framework requiring prior distributions to be specified for all model parameters. Accordingly, we specified minimally informative prior distributions corresponding to a normal (0, 10,000) prior distribution for the pooled mean effects relative to usual care. They are set in a Bayesian framework to allow flexibility in fitting and assigning prior distributions to the parameters of interest while fully accounting for parameter uncertainty. The procedure “mvmeta” of STATA was used for the NMAs. Missing data were handled by augmenting the trial with (mean × 0.001) for effect size and (mean × 10,000 for variance) and between study correlations were set at 0.5. OpenBUGS were used for Bayesian NMAs. All results pertain to 500,000 iterations (10,000 samples for each) and thinning of 100 after a 10,000 burn-in period by using double chains.

As for the risk of bias across studies, publication bias was examined by funnel plot and, when necessary, we conducted sensitivity analyses to adjust for it using the trim and fill method [17,18]. The surface under the cumulative ranking curve (SUCRA) [19] was also determined to examine the mean rank of the interventions. Sensitivity analyses were conducted by using full follow-up duration data, including the Mediterranean diet as an independent intervention.

## 3. Results

Studies were selected by following PRISMA guidelines [20]. Figure 1 presents a flow diagram showing the procedure for selecting studies for our systematic review. 

Appendix A summarizes basic information for each trial. The systematic literature review identified 40 trials that had suitable inclusion and exclusion criteria. Study durations varied from 0.5 to 6 years, and the most frequent duration was 1 year or less [21,22,23,24,25,26,27,28,29]. There were some instances where several papers had been published for one trial. When multiple papers for a trial had been published, we used the paper that reported the results at 1 year (or the results nearest to 1 year) from randomization. We were able to obtain the results at 1 year for half of the studies. Among the 40 eligible studies, three three-arm trials and two four-arm trials were identified by the systematic review. Forty-seven interventions (*n* = 20,113) [21,22,23,24,25,26,27,28,29,30,31,32,33,34,35,36,37,38,39,40,41,42,43,44,45,46,47,48,49,50,51,52,53,54,55,56,57,58,59] were used for the analyses. Seven trials included patients with overweight or obesity and six trials [25,41,51,54,58,59] not only included those with IGT but also with IGT or IFG.

### 3.1. Type of Intervention

The meta-analyses were performed to estimate the odds ratio of interventions. The type of interventions (treatment) in each trial is shown in Appendix A. Lifestyle modification (L) was studied in the largest number of studies (20 trials) followed by metformin (M) (six trials) and dietary (D) and exercise (E) (four trials).

### 3.2. Risk of Bias

We summarized the results of our assessment of the risk of bias for the included studies (Appendix A). All study designs were RCTs and had similar characteristics at baseline. A high level of bias was not found in the design of any of the studies. However, concealment of allocation was difficult to assess in 11 studies [23,25,26,27,28,30,51,52,53,55,56] because of poor reporting, and high risk of bias were confirmed in four studies [32,37,39,42]. Blinding of participants and researchers presented a low risk of bias in 10 studies [21,34,41,42,46,47,48,49,50,57], but blinding of outcome assessment was difficult to assess in three studies [23,37,40]. As for incomplete outcomes, two studies [21,32] had high risk of bias. The overall risk of bias for each of the studies was judged to be high when the quality of the report was low or used the subgroup data. Furthermore, the proportion of each gender varied. The subgroup was treated as an independent study in three studies [32,38,54]. For other biases, it was unclear if other biases were present due to limited information available on the studies. As for the control, we used two treatment types, namely, placebo was one control group and standard intervention was another control group. Although it was apparent that all studies maintained a similar control group and an intervention group, and no results were missing from the final reports, we could not deny the risk of bias based on variability of the quality of the control.

### 3.3. Odds Ratio with Onset of Type 2 Diabetes

Results of the UMA and NMAs (*n* = 40) are shown in Table 1. Heterogeneity, which was examined for the NMA by the inconsistency model of “mvmeta”, was denied (*p* = 0.606). Results of the NMA via the frequentist model are shown in the network map (Figure 2), in which the size of each node and the thickness of the lines are proportional to the number of studies reporting the treatments. The number on each line shows the number of the studies for the direct comparison between the treatments connected by the line.

The odds ratio with onset of type 2 diabetes in the lifestyle modification intervention group was approximately 0.60 (95% CI: 0.48 to 0.76) times lower compared to the standard intervention group by the frequentist approach (random-effects model) and 0.46 (95% CI: 0.33 to 0.61) by the Bayesian approach. Those effects compared with the placebo were also significant. The effects of lifestyle modification compared with other treatments by indirect comparisons varied by the treatments and were not significant. Superiorities of diet, exercise, metformin, acarbose, and orlistat to the standard and placebo treatments were also clarified (see Table 1). The results of the sensitivity analyses for lifestyle modification were not largely different from the main analysis (Appendix A).

The SUCRA was also conducted to examine the mean rank of the interventions (Table 2). Among the treatments, the trials that were not classified as having the worst rank (0.0%), a lower mean rank (<5.0), and a higher SUCRA (≥0.7) were exercise and lifestyle modification interventions. The lifestyle modification as well as exercise, orlistat, and glipizide had a comparatively higher rank by the SUCRA in terms of effectiveness.

As for publication bias, although the funnel plot (Appendix A) indicated that a few published studies had a small sample size, the test for publication bias was not statistically significant, and the possibility of bias could be denied, we conducted sensitivity analyses to adjust for publication bias using the trim and fill method. From the result, it was revealed that the effect of lifestyle modification was still significant (*p* < 0.001) (Appendix A). 

## 4. Discussion

This meta-analysis provides evidence of the efficacy of lifestyle modification in preventing the onset of type 2 diabetes in high-risk patients in comparison with standard treatment or placebo as well as other treatments. The proportion of patients with onset of type 2 diabetes in the intervention group was approximately twice as great compared to the control groups. Results of indirect comparisons of the effects of the lifestyle modification to the other treatments were varied and not significant. However, compared to the effects of the standard or placebo interventions, the lifestyle modification showed stronger effects and the results of SUCRA supported this.

### 4.1. Findings in the Context of the Literature

A former meta-analysis [3] showed that a lifestyle modification intervention could reduce the risk of type 2 diabetes in people with impaired glucose tolerance and was at least as effective as medications. For instance, Gillies et al. [3] reported that, according to a meta-analysis, the estimated hazard ratios with 95%CI were 0.67 (0.49 to 0.92) for diet (D), 0.49 (0.32 to 0.74) for exercise (E), and 0.51 (0.44 to 0.60) for diet and exercise (L). By the Cochran review, the effects of diet and physical activity on the comparator (S) was reported as a relative risk (RR) = 0.57 (0.50 to 0.64) [60]. Our analysis by NMA via the Bayesian approach showed close values, except for diet, which was not significant for pairwise comparison and frequentist NMA. Gillies et al. also analyzed oral diabetes drugs ((A), (M), (G)), an anti-obesity drug (O), and herbal medications (H) compared to placebo (P) and the results showed significant effects for the former two medications. Our results showed significant effects by NMA using the Bayesian approach only for (M). When compared to standard (S), both (M) and (A) were significant but, in contrast, (O) was not. The associations were similar among these studies. Further, a recent meta-analysis of lifestyle modification and use of medication [4] showed that both lifestyle modification and medications (weight loss and insulin-sensitizing agents) successfully reduced the incidence of diabetes, although the effects of medication on the incidence of diabetes were short lived. Further, in their meta-analysis, physical activity was not significant in the subgroup analysis by RR (0.45: 0.11 to 1.82), while our analysis showed significant effects for (E) compared to both standard therapy (0.48: 0.28 to 0.81) and placebo (indirect) (0.38: 0.16 to 0.88) in the NMA Bayesian approach. The reason for this might be that our study included four trials and two newly published papers were added compared to Haw’s paper [4]. As for the herbal medication Tianqi capsule, based on the results of a meta-analysis of six trials by Pang et al. [7], it was reported that there was no statistical difference compared to placebo [P] (including a trial which compared to [L]) by RR (0.89: 0.71 to 1.12). Our NMA Bayesian approach also resulted in no statistical significance (0.51: 0.24 to 1.06), though only four trials were used for the analysis because most of the papers on the topic were written in Chinese. 

Kolb and Martin [2] mentioned that strategies for diabetes prevention should aim at promoting a ’diabetes-protective lifestyle’ whilst simultaneously enhancing the resistance of the human organism to pro-diabetic environmental and lifestyle factors. Comparing drug treatments, lifestyle modification would be expected to incur fewer and less serious side effects and be free from adverse events. A review by Yeung and Mazzola [5] reported that lifestyle modification, mainly diet and physical activity, could be recommended as the best course. In our NMA, treating with both lifestyle modification and several medications simultaneously strengthened the evidence of previous studies. The effects of interventions using drugs may not be permanent while lifestyle modifications may be expected to be followed for longer periods, although the modifications must be done on a regular basis. Furthermore, a recent umbrella review examined the effect of the consumption of a Dietary Approaches to Stop Hypertension (DASH) dietary pattern and diabetes incidence as one of the cardiometabolic outcomes [61] and the authors concluded that the relationship remained uncertain and that future studies would likely have to consider the important influence on risk estimates. In both cases, in the real world more research on diabetes-protective mechanisms [62] as well as the cost of treatments seems warranted.

Currently, various policy initiatives have been suggested to prevent type 2 diabetes [63,64,65]. For instance, the Centers for Disease Control and Prevention (CDC) established the National Diabetes Prevention Program (National DPP) lifestyle change program, which focuses on helping participants make positive lifestyle changes, such as eating healthier and getting more physical activity, in order to address the growing problems of prediabetes and type 2 diabetes [64].

At the independent patient level, a coordinated approach that integrates the beneficial effects of education and support by health care professionals is important. From our analyses, the effects of lifestyle modification in comparison with other treatments proved to be effective. The results of the NMA estimates by frequentist and Bayesian approaches were not largely different, especially for lifestyle modification, on the prevention of type 2 diabetes.

### 4.2. Strengths and Limitations

To our knowledge, this is the first study employing a meta-analysis of RCT studies that has examined the comparative effects of lifestyle modification, dietary supplements, exercise, herbal medication, and several antihyperglycemic medications for individuals at high risk of type 2 diabetes. Previous meta-analyses showed results that were somewhat similar to ours. However, those meta-analyses treated the lifestyle and medication treatments independently or as subgroups of treatments. 

The educational training used was not uniform across reports. Furthermore, we could not distinguish the effects of several lifestyle interventions (such as types of dietary interventions and exercise) and various dosages of medications. The strengths of our study were that we analyzed only RCTs and assessed the magnitude of the effects of the interventions according to the odds ratio with onset of type 2 diabetes primarily using NMA methods. 

We collected data only from studies with a follow-up period extending for more than 6 months. This may be acceptable, however, because earlier assessments could be biased as a result of changes made only because subjects were conscious of being studied. When a trial published more than one paper, we selected the paper that described the result at 1 year or the paper that presented the results closest to 1 year. This may reduce duration bias. However, the bias may be small. By consulting meta-analyses for diet and physical interventions in the Cochran review [60], when the trials were classified into two subgroups according to duration of 4 years or more or less than 4 years, relatively similar RRs were obtained (0.55 vs. 0.57). As sensitivity analyses, we added the analyses using full year follow-up data and treated the Mediterranean diet as an independent intervention; however, the results were not largely different from the results of the 1-year follow-up. In addition, we performed subgroup analysis by classifying study participants into two groups based on the baseline body mass index (BMI), according to those with less than 30 kg/m and those with baseline BMI equal or greater than 30 kg/m. Because of the small sample size, there were not many comparable variables and the comparison was limited to "lifestyle modification vs. standard"; as a result, similar significant odds ratios were obtained. In the prevention of type 2 diabetes, maintaining long-term control is likely important. The results suggested that lifestyle modifications determined by indirect comparisons with the other treatments were effective although not significant and, in addition to conventional education, were more likely to prevent the onset of type 2 diabetes than some other strategies. For the variability as well as quality in lifestyle modification programs, modification was undoubtedly not uniform. In most trials, the control diet and exercise were the subjects’ usual ones, while lifestyle education included some special diets such as the Mediterranean diet [52] among others. Further, most of the studies included recommendations for general exercise. Indeed, considering that the quality of studies of lifestyle, dietary, and exercise modifications may be affected by many confounding biases, these limitations may be acceptable. From a statistical point of view, considering the heterogeneity, we used the random-effects model as the primary analysis. Although the quality and content of lifestyle education varied, the results indicated that it was effective. Because the number of studies was too small to perform difference-by-subgroup analyses, we could not conduct these analyses according to more detailed intervention styles. Taking these limitations into account, this meta-analysis provides evidence of the benefits of long-term regular lifestyle education for reducing the incidence of type 2 diabetes.

Furthermore, the funnel plots denoted weak but non-significant publication bias. To address this, we performed a sensitivity analysis using the trim and fill method. The results showed that there were significant effects of lifestyle modification compared with the standard intervention. Because of this heterogeneity, we used the random-effects model as the primary analysis. Although the quality and content of lifestyle modification varied, the results indicated that it was effective. Because the number of studies was too small to perform difference-by-subgroup analyses, we could not conduct these analyses according to types of intervention. 

We assumed consistency in the model because heterogeneity was not statistically denied. However, if interactions were examined by an inconsistency model for NMA, the effects of the interaction according to the variety of the participants’ characteristics for each trial may be meaningful. Because of these limitations, further study is warranted. Even though our search method involved a systematic review with the addition of hand searching, we could have inadvertently missed eligible studies. In the identification process, we implemented a search strategy that considered other sources (hand searched for information from referring papers and Google Scholar). As for the transparency and transferability of this search strategy, we could not deny the risk of bias across studies. As such, the results should be interpreted carefully when considering the risk of bias across studies.

### 4.3. Implications for Practice and Research

This study evaluated via NMA whether lifestyle modification is effective compared to other treatments of participants at high risk of diabetes. As the results of the present study showed, future studies are suggested in order to clarify which aspects and what degree of lifestyle modifications are best to prevent type 2 diabetes in real life settings.

## 5. Conclusions

This meta-analysis further strengthens the evidence that lifestyle modification is the superior treatment intervention among 12 treatments for the prevention of type 2 diabetes in high-risk individuals, as evaluated by several NMA models. In order to implement lifestyle modification, compliance is the key issue to success. It is recommended that the research should be conducted on the use of policy making for the prevention of diabetes via the creation of programs that support the maintenance and promotion of lifestyle modification.

## Figures and Tables

**Figure 1 nutrients-11-01373-f001:**
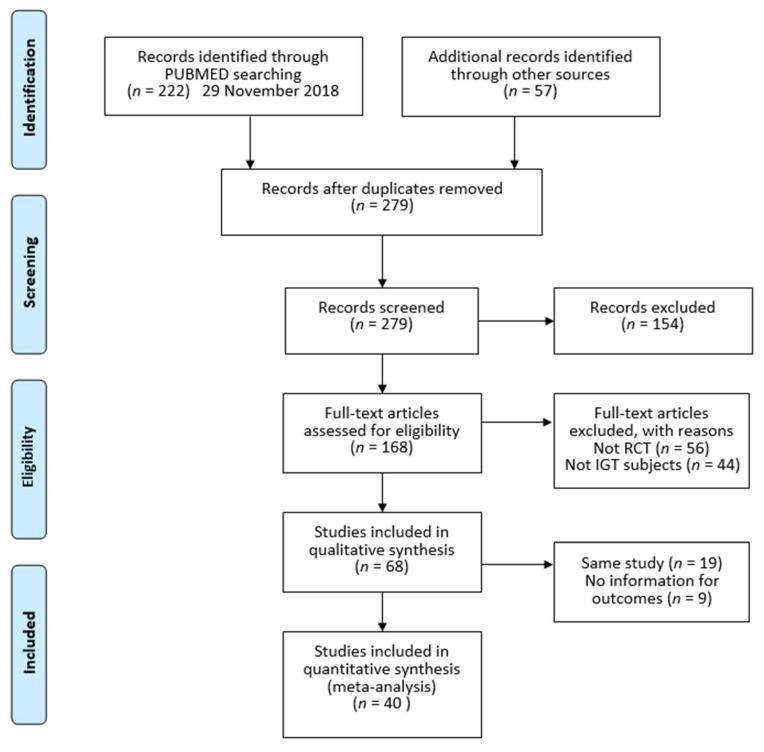
Flow diagram. RCT: randomized controlled trial, IGT: impaired glucose tolerance.

**Figure 2 nutrients-11-01373-f002:**
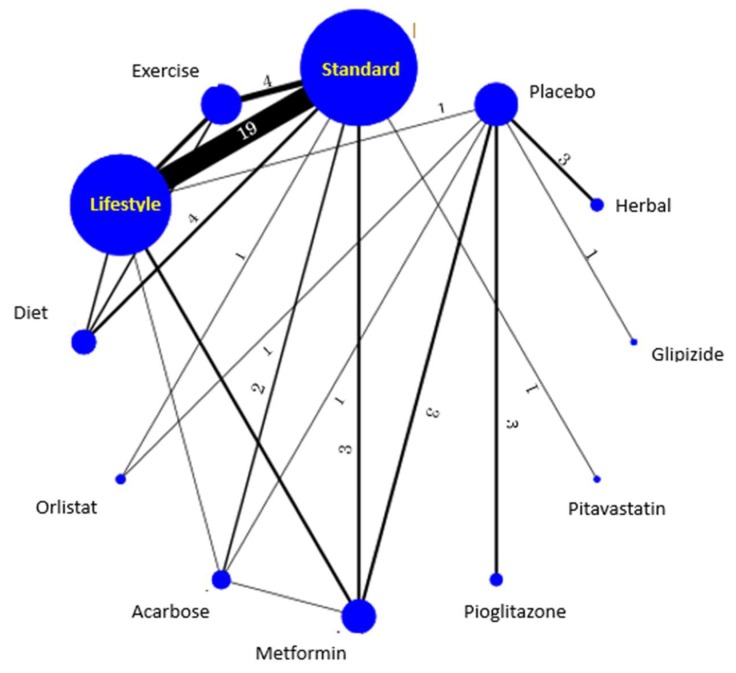
Network map of the 12 treatments by the frequentist model. The size of each node and the thickness of the lines are proportional to the number of studies reporting the treatments.

**Table 1 nutrients-11-01373-t001:** Pairwise and NMA: the estimated parameters by frequentist approach and Bayesian approach.

Interventions		Pair-Wise		NMA (Frequentist Approach) (r = 0.5)	NMA (Bayesian Approach) ^$^
	*n* ^#^	OR (95% CI)	I^2^	OR (95% CI)	OR (95% CI)
**Comparison for Lifestyle**					
Lifestyle vs. Exercise (indirect)				1.18 (0.74,1.93)	1.57 (0.81,3.28)
Lifestyle vs. Diet (indirect)				0.93 (0.60,1.43)	1.01 (0.53,1.94)
Lifestylevs. Orlistat (indirect)				1.19 (0.55,2.58)	1.01 (0.37,2.77)
Lifestyle vs. Acarbose/Voglibose (indirect)				0.92 (0.57,1.48)	0.89 (0.44,1.84)
Lifestyle vs. Metformin/Flumamine (indirect)				0.86 (0.60,1.25)	0.96 (0.58,1.73)
Lifestyle vs. Pioglitazone, Rosiglitazone (indirect)				0.57 (0.32,1.00)	0.57 (0.26,1.36)
Lifestyle vs. Pitavastatin (indirect)				0.68 (0.36,1.29)	0.53 (0.20,1.37)
Lifestyle vs. Glipizide (indirect)				2.07 (0.24,17.7)	3.21 (0.31,99.3)
Lifestyle vs. Herbal medicine (indirect)				0.71 (0.38,1.36)	0.73 (0.29,1.87)
Lifestyle vs. Standard	19	0.65 (0.56,0.75)	4.7	0.60 (0.48,0.76)	0.46 (0.33,0.61)
Lifestyle vs. Placebo	1	0.44 (0.31,0.61)		0.41 (0.27,0.63)	0.38 (0.20,0.71)
**Comparison for Lifestyle**					
Diet vs. Standard	4	0.71 (0.55,0.90)	35.7	0.65 (0.43,0.98)	0.46 (0.24,0.84)
Exercise vs. Standard	4	0.45 (0.24,0.84)	29.9	0.50 (0.31,0.81)	0.29 (0.14,0.55)
Orlistat vs. Standard	1	0.49 (0.25,0.95)		0.51 (0.24,1.07)	0.45 (0.17,1.19)
Acarbose/Voglibose vs. Standard	2	0.70 (0.31,1.57)	13.4	0.66 (0.42,1.04)	0.51 (0.25,0.99)
Metformin/Flumamine vs. Standard	3	0.40 (0.10,1.68)	75.4	0.70 (0.47,1.03)	0.48 (0.26,0.79)
Pioglitazone, Rosiglitazone vs. Standard (indirect)				1.06 (0.60,1.90)	0.80 (0.33,1.76)
Pitavastatin vs. Standard	1	0.88 (0.65,1.20)		0.88 (0.49,1.59)	0.87 (0.34,2.17)
Glipizide vs. Standard (indirect)				0.29 (0.03,2.51)	0.14 (0.00,1.45)
Herbal medicine vs. Standard (indirect)				0.85 (0.44,1.61)	0.63 (0.24,1.58)
Placebo vs. Standard (indirect)				1.46 (0.95,2.25)	1.23 (0.63,2.23)
**Comparison with Placebo**					
Diet vs. Placebo (indirect)				0.45 (0.25,0.79)	0.37 (0.16,0.89)
Exercise vs. Placebo (indirect)				0.35(0.19,0.64)	0.38 (0.16,0.88)
Orlistat vs. Placebo	1	0.40 (0.07,2.24)		0.35 (0.15,0.79)	0.35 (0.14,0.90)
Acarbose/Voglibose vs. Placebo	1	0.38 (0.25,0.58)		0.45 (0.28,0.74)	0.42 (0.20,0.87)
Metformin/Flumamine vs Placebo	3	0.52 (0.30,0.91)	64.4	0.48 (0.32,0.71)	0.39 (0.21,0.67)
Pioglitazone, Rosiglitazone vs. Placebo	3	0.70 (0.44,1.11)	87.3	0.73 (0.50,1.05)	0.65 (0.37,1.11)
Pitavastatin vs. Placebo (indirect)				0.64 (0.29,1.42)	0.71 (0.24,2.23)
Glipizide vs. Placebo	1	0.22 (0.02,1.90)		0.20 (0.02,1.64)	0.12 (0.00,1.13)
Herbal medicine vs. Placebo	3	0.60 (0.45,0.82)	0.0	0.60 (0.29,1.26)	0.51 (0.25,1.02)
Log Likelihood Ratio/*p*-values for inconsistency				*χ*^2^ = 7.24, *p* = 0.78 (df = 11)	Tau = 0.42 (0.23,0.66)

Univariate and NMA for T2D was analyzed using STATA (mvmeta). For NMA with random effects, Open BUGS was used for the estimation. $: with large variance (10,000), number of iterations: 500,000, number of burn-in period: 10,000, thining: 100. Herbal medicine includes Jiangtang bushen recipe, and Tianqi. #: number of studies used for the direct comparison.

**Table 2 nutrients-11-01373-t002:** Surface under the cumulative ranking curve (SUCRA) by NMA ^$^.

Study rank	Lifestyle	Exercise	Diet	Orlistat	Acarbose/Voglibose	Metformin/Flumamine	Pioglitazone/Rosiglitazone	Pitavastatin	Glipizide	Herbal medicine	Placebo	Standard
Worst	0.0	0.0	0.1	0.4	0.0	0.0	4.1	8.0	6.7	1.3	77.0	2.5
Mean	4.4	3.1	5.4	3.6	5.4	6.0	9.6	8.2	3.0	7.7	11.7	9.5
SUCRA	0.7	0.8	0.6	0.8	0.6	0.5	0.2	0.3	0.8	0.4	0.0	0.2

$: By MVMETA assuming the minimum parameter is the best, using 5000 draws, allowing for parameter uncertainty. Worst (%) denotes the estimated probabilities (%) of each treatment being the worst rank (0%: best to 100%: worst). Mean rank denotes average rank in the 12 treatments for 5000 repeatments (1: best to 12: worst). SUCRA is the surface under cumulative ranking curve (1: best to 0: worst). Bold shows interventions with (Worst = 0%, Mean rank < 5, or SUCRA ≥ 0.7).

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
