# Peer review of "Comparison of the Effectiveness of Lifestyle Modification with Other Treatments on the Incidence of Type 2 Diabetes in People at High Risk: A Network Meta-Analysis"

_nutrients, 2019, doi:10.3390/nu11061373_

Reviewer 1 Report

A good and interesting study. Easy to follow and easy to read. Here are some few comments that the authors can consider improving the study furthermore.

 1.     The inclusion criteria and exclusion criteria are not clear. I suggest using PICO tool in a table with inclusion and exclusion criteria. The table can be in the supplementary materials.

2.     Regarding the supplementary materials, the reference numbers are not in order. I suggest to double check the reference number as well as the names of the Author (Knowler is written as “Knower” for the DPP study).

3.     I would like to see more information on the S1 table. For example, which criteria were used to judge high risk group, i.e. obesity or IGT etc.

4.     I am not sure why the authors choose to use study follow-up close to 1 year. For example, the DPP study duration was 2.8 years whereas the DPS duration was 3 years.

5.     Regarding my previous comment, I would also like to see a column in S1 table mentioning the actual intervention duration. The DPS intervention was 3 years. Please provide this information for all the studies.

6.     Also discuss this issue in the discussion section, how you think it will affect your results.

7.     The risk of bias estimation, I would like to see an overall judgment for each study according to the author. If it is not possible to provide an overall judgment, please explain why

8.     In the flow diagram, you mention 57 studies have been included form other sources. Please mention what were the “other sources”. Also, describe your search strategy for other sources for transparency and transferability

9.     Line 147 to 148, please prove the references for the 15 and 11 studies mentioned

10.  Line 224, provide reference for the How’s paper

11.  Line 205, Former meta-analyses (3), you just provide reference for 1 paper.

Author Response

For Reviewer 1:

 Thank you very much for your valuable comments and suggestions concerning our manuscript. We found these most helpful and have revised the manuscript accordingly. Following are answers to the comments and explanations for the major revisions, which are reflected in the manuscript. Revisions were clearly highlighted by using the "Track Changes" function in Microsoft Word.

 We hope that the corrections are satisfactory.

 MAJOR COMPULSORY REVISIONS

 1.     The inclusion criteria and exclusion criteria are not clear. I suggest using PICO tool in a table with inclusion and exclusion criteria. The table can be in the supplementary materials.

Thank you for your suggestion. We added Table S1 using the PICO tool in the supplementary materials.

2.     Regarding the supplementary materials, the reference numbers are not in order. I suggest to double check the reference number as well as the names of the Author (Knowler is written as “Knower” for the DPP study).

Thank you for your careful attention to the reference numbers. We have made double check the reference number and the names of the Author and corrected.

 3.     I would like to see more information on the S1 table. For example, which criteria were used to judge high risk group, i.e. obesity or IGT etc.

5.     Regarding my previous comment, I would also like to see a column in S1 table mentioning the actual intervention duration. The DPS intervention was 3 years. Please provide this information for all the studies.

 We added information related to criteria, actual intervention duration, and other (subgroup or stratified analysis results) on the Table S2 (old Table S1).

 4.     I am not sure why the authors choose to use study follow-up close to 1 year. For example, the DPP study duration was 2.8 years whereas the DPS duration was 3 years.

6.     Also discuss this issue in the discussion section, how you think it will affect your results.

As described in the limitation, the reason why we selected result at 1 year or nearby was to reduce duration bias. The duration of the studies used in this review varied from 6 months to 6 years. The most frequent duration was 1 year or less (23, 30, 32, 43, 46, 50, 56, 57, 58) and we could have obtained the results at 1 year among a half of the studies. We thought that the bias may be small. However, considering to maintain long-term control be warranted in the prevention of T2D, we added the analysis using the full study follow-up result and showed it in the Supplementary Table S4 (in the analyses, we used Mediterranean diet as an independent intervention following Reviewr2’ suggestion). The result was not largely different from the main result except for the Metformin and Mediterranean diet. Namely those became significant not in the 1 year, but in the full follow-up-years. We added this in the discussion section.

We added the following sentence in “Data extraction and risk of bias within individual studies” section as well as “Discussion” section.

 Data extraction and risk of bias within individual studies

Line 94-97: “Although considering to maintain long-term control be warranted in the prevention of T2D, the reason why we selected result at 1-year study follow-up duration was to reduce duration bias Study follow-up durations.”

 In Results section,

Line 131-132: “and the most frequent duration was 1 year or less (23, 30, 32, 43, 46, 50, 56, 57, 58).”

 Line 134: “We could have obtained the data at 1 year among a half of the studies.”  

 In the Discussion section,

Line 278-283: As a sensitivity analysis, we added the analyses by using full follow-up duration data, including the Mediterranean diet as an independent intervention, the results were not largely different from the results of at 1-year follow-up.

 7.     The risk of bias estimation, I would like to see an overall judgment for each study according to the author. If it is not possible to provide an overall judgment, please explain why

An overall judgement (High, Middle, or Low) for each study was added in Figure S1.  The overall judgment of the low was made based on the low quality of the report including subgroup data. Lifestyle intervention was a so-called "complex intervention" and we concerned that the intervention was not implemented as planned. We did not place too much emphasis on the risk of other biases. For example, although two items (random sequence generation and allocation concealment) generally affect selection bias, we did not consider them to be critical in the overall judgement for this review because safety concerns for interventions were relatively low.

We added the following sentence in “Risk of bias” of the Results session.

 Line 148-157: 3.2. Risk of bias

“We summarized results of our assessment of the risk of bias for the included studies (Supplementary Figure S1). All study designs were RCTs and had similar characteristics at baseline.  A high level of bias was not found in the design of any of the studies. However, concealment of allocation was difficult to assess in 11 studies [21, 31, 46, 47, 48, 49, 50, 52, 53, 56, 57] because of poor reporting, and high risk of bias were confirmed in 4 studies [24, 30, 33, 36]. Blinding of participants and researchers presented a low risk of bias in 10 studies [23, 26, 35, 36, 40, 41, 42, 44, 45, 54] but blinding of outcome assessment was difficult to assess in 3 studies [34, 30, 31]. As for the incomplete outcome, 2 studies [23,24] were high risk of bias. The overall risk of bias for each studies was judged to be high when the quality of the report was low or used the subgroup data.”

8.     In the flow diagram, you mention 57 studies have been included form other sources. Please mention what were the “other sources”. Also, describe your search strategy for other sources for transparency and transferability

The “other sources” were hand searched for the information about referring papers and google scholar. As for the transparency and transferability of this search strategy, we could not deny the risk of bias across studies. However, in terms of searching as widely as possible, we thought it should make a sense. Note that for the publication bias, we examined funnel plot as well as the trim and fill method.

To make it clear, we stated this in Strengths and limitations of the study of the discussion section.

Line 311-314: “In the identification process, we used the “other sources” (hand searched for the information about referring papers and google scholar). As for the transparency and transferability of this search strategy, we could not deny the risk of bias across studies. So, the results should be interpreted carefully considering the risk of bias across studies.”

Minor issue:

9. Line 147 to 148, please prove the references for the 15 and 11 studies mentioned

We added the reference numbers accordingly.

10.  Line 224, provide reference for the How’s paper

We added the reference number.

11.  Line 205, Former meta-analyses (3), you just provide reference for 1 paper.

We corrected “meta-analyses” to “meta-analysis”.

Reviewer 2 Report

- This is important but not very novel analysis. Results and groups are too superficial such as: 

What is the details of life style? Details of standard intervention?

Diet resulted in weight loss? or diet resulted in weight gain or neutral? Any difference between the end points? For example Mediterranean diet often does not result with weight loss, however high quality fat still has huge effect on metabolic health. If there is caloric restriction in diet, would group them separately to see how intensive restriction you need for DM prevention.

What is the intensity of exercise? Is there a difference in terms of DM prevention with low intensity vs. high intensity exercise?

If authors can perform those more detailed analysis, it would be great contribution to recent literature, otherwise it is all known and nobody really challenges the effectiveness of lifestyle intervention.

For minor English edits:

38. I would delete etc. , which is not a good scientific term. 

78 and other parts, I would use the word medication rather than medicine.

214. You wrote out, likely meant our.

Author Response

For Review 2:

Thank you very much for your valuable comments concerning our manuscript. We found these most helpful and have revised the manuscript accordingly. Following are answers to the comments and explanations for the major revisions, which are reflected in the manuscript. Revisions were clearly highlighted by using the "Track Changes" function in Microsoft Word.

 We hope that the corrections are satisfactory.

Major issues:

- This is important but not very novel analysis. Results and groups are too superficial such as: 

What is the details of life style? Details of standard intervention?

Diet resulted in weight loss? or diet resulted in weight gain or neutral? Any difference between the end points? For example, Mediterranean diet often does not result with weight loss, however high quality fat still has huge effect on metabolic health. If there is caloric restriction in diet, would group them separately to see how intensive restriction you need for DM prevention.

What is the intensity of exercise? Is there a difference in terms of DM prevention with low intensity vs. high intensity exercise?

If authors can perform those more detailed analysis, it would be great contribution to recent literature, otherwise it is all known and nobody really challenges the effectiveness of lifestyle intervention.

Thank you for your critical comments. We agree that those points are very important.

To examine the effects on special risk factors on DM such as weight and cholesterol levels independently is important issue. However, onset of DM is primary endpoint and, for this viewpoint, we think this study could have some significance on DM prevention studies. We think the analyses for the specific risk factors as a primary endpoint should be done in the future.

We agree that the Mediterranean diet is considered having a beneficial effect on obesity, we added sensitivity analyses for dealing the Mediterranean diet as an independent treatment. Furthermore, for the obesity, we added additional analysis by classified the studies into two groups based on the baseline BMI level (BMI<30, bmi="">=30). The results are shown in Supplementary Table S4.

Although the number of studies used in the analyses became small and confidence intervals became wide, our results suggested that not only the Mediterranean diet but also the other dietary modification can be beneficial though not statistically significant. For these points, we added the following sentences in the discussion section.

Line 278-283: “As a sensitivity analysis, we added the analyses by using full follow-up duration data, including the Mediterranean diet as an independent intervention, the results were not largely different from the results of at 1-year follow-up. In addition, we performed subgroup analysis by classified into two groups based on the baseline BMI with less than 30kg/m and equal or greater 30kg/m. Because of small sample size, there were not so many comparable variables and it was limited to "lifestyle modification vs. standard", the effects were similar significant odds ratios were obtained.

Line287-298: For the variability as well as quality in lifestyle modification programs, surely modification was not uniform. In most trials, the control diet and exercise were the subject’s usual ones, while lifestyle education included some special diets such as the Mediterranean diet (29) among others. Further, most of the studies included recommendations for general exercise.

Indeed, considering that the quality of studies of lifestyle, dietary, exercise modifications may be affected by many confounding biases, these limitations may be acceptable.

From statistical point of view, considering the heterogeneity, we used the random-effects model as the primary analysis. Although the quality and content of lifestyle education varied, the results indicated that it was effective. Because the number of studies was too small to perform difference-by-subgroup analyses, we could not conduct these analyses by more detailed intervention styles. Taking these limitations into account, this meta-analysis provides evidence of the benefits of long-term regular lifestyle education, for reducing the incidence of type 2 diabetes.

Minor issue:

38. I would delete etc. , which is not a good scientific term. 

We corrected it accordingly.

78 and other parts, I would use the word medication rather than medicine.

We corrected “medicine” into “medication” accordingly throughout the manuscript.

214. You wrote out, likely meant our.

We have made this correction. Thank you.

Round  2

Reviewer 2 Report

I think manuscript is significantly improved and their responses are satisfactory.